# From Molecular to Clinical Implications of Sleep-Related Breathing Disorders on the Treatment and Recovery of Acute Stroke: A Scoping Review

**DOI:** 10.3390/cimb47030138

**Published:** 2025-02-21

**Authors:** Karol Uscamaita, Olga Parra Ordaz, Imán Yazbeck Morell, Marta García Pla, María-José Sánchez-López, Adrià Arboix

**Affiliations:** 1Neurology Service, Sleep Disorders Unit, Hospital Universitari Sagrat Cor, Grupo Quirónsalud, 08029 Barcelona, Spain; karol.uscamaita@quironsalud.es; 2Medicine Department, University of Barcelona, 08036 Barcelona, Spain; oparra@ub.edu; 3Department of Pneumology, Sleep Disorders Unit, Hospital Universitari Sagrat Cor, Grupo Quirónsalud, 08029 Barcelona, Spain; 4Department of Internal Medicine, Hospital Universitari Sagrat Cor, Grupo Quirónsalud, 08029 Barcelona, Spain; iman.yazbeck@quironsalud.es; 5Emergency Department, Hospital Universitari Sagrat Cor, Grupo Quirónsalud, 08029 Barcelona, Spain; martagarcia.bcn.ics@gencat.cat; 6Medical Library, Hospital Universitari Sagrat Cor, Grupo Quirónsalud, 08029 Barcelona, Spain; biblioteca.hsc@quironsalud.es; 7Cerebrovascular Division, Department of Neurology, Hospital Universitari Sagrat Cor, Grupo Quirónsalud, Universitat de Barcelona, 08029 Barcelona, Spain

**Keywords:** stroke, ischemic stroke, CPAP, pathophysiology, sleep apnea

## Abstract

(1) Background: The aim of this review is to map research into the molecular mechanisms linking sleep-related breathing disorders (SRBDs) and acute stroke and their clinical and therapeutic implications and to identify existing knowledge gaps to suggest new areas of research. (2) Methods: This review was conducted according to the PRISMA extension for scoping reviews (PRISMA-ScR) and a predetermined protocol shared among all authors. (3) Results: The review of the thirteen studies analyzed provides a focused view of the molecular features about interaction between obstructive sleep apnea (OSA) and acute stroke. Our review identifies and highlights the biomarkers most frequently found to be associated with acute stroke, SRBDs, and their clinical repercussions. (4) Conclusions: The association between the presence of sleep apnea, especially in its severe form, and elevated levels of inflammatory markers in patients with acute stroke is highlighted and new research topics in this area are proposed.

## 1. Introduction

Sleep-related breathing disorders (SRBD) are characterized by abnormalities of respiration during sleep. These disorders include obstructive sleep apnea (OSA) disorders, central sleep apnea disorders, sleep-related hypoventilation disorders, and sleep related hypoxemia disorder [1]. OSA are clearly the most frequent SRBD in general population, affecting between 23 to 49% of general population [2].

OSA is characterized by repetitive episodes of complete (apnea) or partial (hypopnea) upper airway obstruction occurring during sleep. These events often result in reductions in blood oxyhemoglobin saturation and are usually terminated by brief arousals from sleep. Oxyhemoglobin saturation usually returns to baseline values following resumption of normal breathing but may remain low if the apneic or hypopneic events are very frequent and prolonged, or if there is underlying pulmonary pathology [1].

Worldwide, stroke continues to be leading cause of death in women, the second most common cause of mortality, and the second most common cause of disability in the adult population [3]. Specifically in Spain, according to data extracted from the Annual Report of the National Health System 2020–2021, cerebrovascular disease affects 1.5% of the population, although it affects six out of every hundred people over the age of 65 and ten out of every hundred over the age of 80 [4,5]. While the incidence of stroke and overall stroke-related mortality rate is decreasing, the absolute number of people with stroke, stroke survivors, stroke-related deaths, and the overall burden of stroke-related disability is high and increasing [6,7,8].

Strokes can be classified into ischemic or hemorrhagic. Ischemic strokes are episodes of neurological dysfunction caused by focal cerebral, spinal, or retinal infarction. Hemorrhagic strokes are focal collections of blood within the brain parenchyma or ventricular system that are not caused by trauma [8]. An acute stroke is a heterogeneous disease, and in clinical studies, it is extremely important to adequately differentiate between the different stroke subtypes (atherothrombotic infarct, cardioembolic stroke, lacunar infarct, infarct of unusual etiology, essential cerebral infarct, intracerebral hemorrhages) among the population with stroke. This recommendation is due to the impact of the stroke subtypes on the distribution of risk factors, stroke severity, and outcome [9].

SRBDs, particularly obstructive sleep apnea (OSA), are frequent in the acute phase of stroke, with approximately 62% of patients experiencing elevated levels of sleep apnea within the first 24 h of stroke onset [10,11,12,13,14,15]. The presence of SRBDs in acute stroke is highly relevant. Over the last 15 years, we have witnessed a growing interest in the relationship between SRBDs and stroke. Several randomized controlled trials demonstrated that sleep-related breathing disorders are linked to early neurological worsening after stroke [16,17,18,19,20,21,22] and an increase in its mortality [23,24,25,26,27,28].

Some studies have provided pathophysiological information on how SRBDs, especially OSA, worsen the prognosis in acute stroke. It has been demonstrated that during periods of apnea, heart rate decreases; although the specific mechanism is unknown, it is believed to be due to incomplete autonomic injury caused by the intermittent hypoxia. On the other hand, it has also been shown that there is lower cerebral blood flow during episodes of obstructive hypopnea. This has been attributed to the fact that during hypopnea episodes, negative intrathoracic pressure occurs in an attempt to overcome the obstruction of the upper airway. This would increase the heart’s afterload and decrease cerebral blood flow. It is also theorized that this may be due to a reflex vasodilation caused by the hypoxia produced by each apnea episode [19,20].

Clinical trials in the general population have showed that treatment of SRBDs with continuous positive airway pressure (CPAP) or other therapies significantly reduces oxyhemoglobin desaturations and apnea. Moreover, a large number of biomarkers related to sympathetic activation, endothelial dysfunction, hypercoagulability, oxidative stress, inflammation, and metabolic dysregulation have been studied in patients with SRBDs, as these are some of the major mechanisms linking SRBDs to cardiovascular disease. However, to date, limited evidence is available on the efficacy of CPAP in stroke patients due to unclear strategy and timing [29,30,31].

To date, screening SRBDs or the use of CPAP devices are not indicated in acute stroke [24], as various studies have failed to demonstrate its relevance in improving the functional status of patients or reducing mortality [32,33,34,35]. The aim of this review is to systematically map research on the molecular mechanisms linking SRBD and acute stroke, their clinical and therapeutic implications, and to identify existing knowledge gaps to suggest new areas of research.

## 2. Materials and Methods

This review was conducted according to the PRISMA extension for scoping reviews (PRISMA-ScR) [28,29] and a predetermined protocol shared among all authors. Institutional review board approval was no necessary as this study utilized only published data. Study protocol was register in osf.io with DOI: 10.17605/OSF.IO/375KD.

Eligibility criteria

Studies assessing molecular, clinical, and therapeutic repercussions of SRBDs on acute stroke as primary or secondary outcome variables were selected. Only published articles up to December 2023, written in English, Spanish, French, or German, involving human participants were included. The study design included quantitative, qualitative and mixed-methods studies to consider a broad catchment of the topic. Letters to the editor, editorials, commentaries, published protocols, conference proceedings, studies only published in abstract form, and case reports were excluded. No context limit was applied.

Information sources

The PubMed, Scopus, and Web of Science databases for studies published from inception to 31 December 2023 in English, Spanish, French, or German, were searched. The search query was constructed in PubMed based on key terms “sleep apnea syndrome” and “stroke” combined with key term “biomarkers” and a set of terms that refer to different biomarkers options. Th search was adapted for use in the other databases [Appendix A]. Studies not published in journals, and articles in languages other than those selected were excluded.

Study Selection and Data Extraction

The title/abstract screening was conducted by two authors of the study. Subsequently, two different authors independently reviewed the eligibility of the studies included and extracted data. Full text articles were assessed by two of the authors and disagreement was resolved by consulting a third author. The final selection was shared among all the six coauthors and any discrepancies were resolved through discussion and consensus.

Data items

Information from each included article was summarized using a structured form, including data on article characteristics (study design), settings, detailing the summary of findings and key outcomes.

## 3. Results

Our scoping review identified 186 articles and, following the removal of 25 duplicated records, we included a total of 162 studies for screening. From these, 60 were found irrelevant, and the remaining 102 were screened based on title and abstract. Of those, a total of 61 studies were excluded because they did not meet the inclusion criteria.

Finally, 41 full-text studies were reviewed. Among these, 24 studies were excluded for not reporting on the topic of the study (n = 24), conference proceedings and letter to the editor (n = 3), and non-English language (n = 1), resulting in 13 studies considered eligible (Figure 1).

The included studies were Ifergane et al., 2016 [30]; Chen et al., 2015 [31], von Känel et al., 2013 [32]; Jensen et al., 2018 [33]; Yeh et al., 2017 [34]; Lévy et al., 2009 [35]; Gregori-Pla et al., 2018 [36]; Tsai et al., 2010 [37]; Fei et al., 2021 [38] Mashaqi et al., 2021 [39]; Kunz et al., [40]; Azarbarzin et al., 2012 [41]; and Medeiros et al., 2012 [42].

Of the 13 studies reviewed, 7 (53.8%) were prospective observational studies, mostly cross-sectional, 4 (30.7%) were reviews of other studies, and 2 (15.3%) were prospective clinical trials. Almost all studies investigate the molecular relationships of ischemic stroke with sleep-related breathing disorders (SRBDs), with only one focusing on hemorrhagic stroke.

In 7 of the 13 reviewed studies [30,32,35,38,39,40,42], an elevation of pro-inflammatory factors was demonstrated, among which interleukin 6 (IL-6) was found elevated most frequently; there is also significant evidence regarding the elevation of matrix metalloproteinase-9. Other pro-inflammatory factors that have been found elevated include C-reactive protein (CRP), caspase-3, Bax, cytokines IL-1β, TNF-α, NF-κB, and soluble tumor necrosis factor receptors -1 and -2. One study highlights that elevated levels of soluble intercellular adhesion molecule-1 (sICAM-1) are related to sleep fragmentation after stroke rehabilitation, serving as an inflammatory marker that links functional improvement from stroke to sleep quality [34].

One of the studies focused on investigating the variation of prothrombotic factors in patients with SRBD treated with CPAP and without CPAP; this study found no differences in these biomarkers [32].

Two studies focused on measuring cerebral blood flow in patients with sleep apnea using different methodologies; both concluded that patients with obstructive sleep apnea (OSA) have lower cerebral blood flow during apneas and exhibit altered vasoreactivity that correlates with the severity of OSA [33,36].

Details of each study are included in Table 1.

## 4. Discussion

The relationship between sleep-related breathing disorders (SRBDs) and stroke has gained significant attention in recent decades, with research beginning in the 1990s [12,43]. Over the past 30 years, numerous studies have explored the connections between various types of stroke (both ischemic and hemorrhagic) in acute, chronic, and preventive phases. Our review focused specifically on molecular alterations in patients in the acute phase of stroke (both ischemic and hemorrhagic) with SRBD and its clinical repercussions, limiting our analysis to 13 studies. The heterogeneity of these studies precludes a systematic review that could combine results and highlight biases in each study.

Regarding acute hemorrhagic stroke, the interplay between obstructive SRBD and intracerebral hemorrhage (ICH) presents a complex pathophysiological scenario with significant clinical implications. Recent studies have elucidated the molecular mechanisms underlying this relationship, focusing on apoptotic mediators and inflammatory cytokines [44,45,46,47,48,49,50]. However, a critical limitation in current animal models is the inability to distinguish between inflammatory markers produced by ICH and those resulting from OSA.

Our review shows that SRBD induction decreases survival rates, neurological scores, and neuronal survival while upregulating protein expression levels of caspase-3, Bax, and pro-inflammatory cytokines such as IL-1β, IL-6, and TNF-α [38]. These findings suggest that OSA-mediated induction of apoptosis and neuroinflammation exacerbates neuronal death following ICH. However, the inflammatory response observed in these models may be confounded by the combined effects of ICH and SRBD, due to the overlap of inflammatory biomarkers produced by both conditions, making it challenging to delineate the specific contribution of each to the overall inflammatory state. This limitation underscores the need for more refined experimental approaches to elucidate the distinct roles of ICH and SRBD in neuroinflammation and subsequent brain injury.

Regarding acute ischemic stroke (AIS), our review also highlights a complex interplay with SRBDs, revealing a shared inflammatory profile. While interleukin-6 (IL-6) is elevated in both conditions [51,52,53,54], several biomarkers appear to be specifically associated with the comorbidity of SRBDs and acute ischemic stroke:Soluble tumor necrosis factor receptors (sTNF-R1 and sTNF-R2)Plasminogen activator inhibitor-1 (PAI-1) [30,40,42]

These SRBD-specific inflammatory markers in AIS patients suggest a unique pathophysiological pathway linking the two conditions. The identification of these biomarkers opens up new possibilities for targeted immunomodulatory therapies, which could potentially mitigate the detrimental effects of SRBDs on acute ischemic stroke treatment outcomes and offer novel approaches to stroke management in patients with comorbid SRBDs.

Serum total antioxidant capacity (TAC) plays a crucial role in acute ischemic stroke by reflecting the overall ability of the body to counteract oxidative stress [31]. Our review found that SRBDs and AIS present a paradoxical scenario in terms of oxidative stress and antioxidant capacity. In patients with severe SRBDs (AHI > 30), elevated TAC levels have been observed, suggesting an adaptive response to chronic intermittent hypoxia. This increase in TAC appears to attenuate the inflammatory reaction typically associated with SRBD in ischemic stroke patients. Conversely, AIS patients generally exhibit lower TAC levels, which correlate with poorer functional outcomes and increased neurological deficits.

This apparent contradiction may explain the exacerbated outcomes observed in patients with comorbid AIS and SRBD. We propose that the cumulative oxidative stress burden from both conditions may overwhelm the compensatory increase in TAC associated with SRBD alone. The insufficient antioxidant capacity to counteract the combined oxidative insult could be a key factor in the poor prognosis of these patients. Therefore, aggressive management of SRBD in the acute stroke setting could be crucial for improving outcomes.

In the context of acute ischemic stroke, MMP-9 is implicated in blood-brain barrier (BBB) disruption and subsequent brain injury. Elevated MMP-9 levels are associated with increased infarct size and worse clinical outcomes due to its role in neuroinflammation and tissue destruction. MMP-9 facilitates the breakdown of the BBB, allowing for the infiltration of inflammatory cells and further exacerbating ischemic damage [52]. In our review, MMP-9 has been implicated in AIS associated with SRBD. Higher MMP-9 levels in SRBD patients are linked to AIS [39], likely due to the hypoxia and oxidative stress characteristic of SRBD. This is a strong link mechanism in both pathologies that deserve deeper research regarding its role in acute stroke and its treatment.

Regarding procoagulant biomarkers, SRBD patients showed lower mean and amplitude of D-dimer and higher mean of PAI-1. There may be altered day/night rhythm of some prothrombotic markers in SRBD. Continuous positive airway pressure (CPAP) treatment did not affect the day/night rhythm of prothrombotic markers [32]. While CPAP therapy has shown efficacy in managing SRBD symptoms, our review suggests that it may not fully address the prothrombotic state associated with this condition. Elevated levels of PAI-1 and D-dimer in SRBD patients indicate a persistent prothrombotic environment, which may contribute to increased stroke risk and poorer outcomes. While CPAP therapy effectively manages respiratory symptoms and improves cardiovascular outcomes in SRBD patients, its impact on these prothrombotic markers remains unclear. This suggests that even with optimal CPAP adherence, SRBD patients may remain at elevated risk for thrombotic events. Future research should focus on developing targeted therapies to address this residual prothrombotic state, potentially combining CPAP with anticoagulant or antifibrinolytic treatments to comprehensively manage stroke risk in SRBD patients.

Regarding cerebral blood flow, alterations and impaired cerebrovascular reactivity in SRBD patients [36], may contribute to increased stroke risk, although some studies show conflicting results. These shared molecular pathways and biomarkers not only enhance our understanding of the SRBD–stroke relationship but also offer potential targets for future diagnostic and therapeutic interventions.

## 5. Strengths and Limitations

The methodology of this review exhibits several key strengths that enhance its credibility and rigor. By following the PRISMA-ScR guidelines, the review ensures a systematic and transparent approach, which is further reinforced by a predetermined protocol shared among all authors. The inclusion of diverse study designs—quantitative, qualitative, and mixed-methods—allows for a comprehensive exploration of the impact of sleep apnea syndrome (SBD) on acute stroke. A meticulous screening process conducted by multiple authors minimizes bias and enhances the quality of included studies. Additionally, utilizing reputable databases like PubMed, Scopus, and Web of Science ensures access to high-quality data. Collectively, these strengths provide a solid foundation for understanding the complex relationship between SBD and acute stroke outcomes.

The study also has limitations, including potential selection bias from excluding letters to the editor and abstracts, which may omit valuable insights. Language restrictions to English, Spanish, French, and German could introduce bias by excluding relevant studies in other languages. The focus on the molecular, clinical, and therapeutic aspects of SRBDs and stroke may overlook other important factors. Additionally, the inclusion of diverse study methodologies could lead to variability in data, complicating comparisons. Publication bias is a concern, as relying solely on published studies may exclude negative results. The lack of a systematic quality assessment for included studies affects reliability, and searching only three databases may have missed relevant research. Finally, the review is limited to studies published until December 2023, leaving out any subsequent findings.

## 6. Conclusions

The relationship between SRBDs and cardiovascular conditions is significant, encompassing stroke (ischemic and hemorrhagic), hypertension, myocardial infarction, post-myocardial infarction heart failure, and atherosclerosis.

Markers including matrix metalloproteinase-9 (MMP-9) and tissue inhibitor of metalloproteinase-1 are prevalent in individuals with obstructive sleep apnea (OSA), hypertension, myocardial infarction, and heart failure.

The existence of severe obstructive sleep apnea in stroke patients may affect the inflammatory response and the severity of the stroke.

The impact of SRBDs on cerebral and cardiovascular responses includes potential alterations in brain response to positional changes and cerebral vasoreactivity, which may influence the regulation of cerebral blood flow.

Identifying inflammatory and oxidative stress markers in patients with OSA may be essential for prognostic prediction and therapeutic guidance.

The influence of SRBDs on inflammatory and cardiovascular response during acute stroke may alter the inflammatory response in acute stroke patients, thereby affecting the severity and prognosis of the cerebrovascular incident.

The paucity of studies investigating the beneficial effects of CPAP in the acute phase of stroke, particularly in relation to its impact on the pathophysiological mechanisms identified in various articles, represents a significant gap in current research. This lack of evidence underscores the need for further investigation into how CPAP intervention might modulate the acute inflammatory and prothrombotic states associated with comorbid OSA and stroke, potentially improving outcomes in this high-risk patient population.

The prompt recognition of sleep apnea in acute stroke patients could be important for a possible therapy and for forecasting potential complications and long-term outcomes.

## 7. Future Research Topics

Future studies should focus on longitudinal designs to establish causality and explore the potential of novel therapeutic approaches targeting the identified molecular pathways.

It is important to highlight that studies with a stronger methodology, such as randomized clinical trials, are needed. Based on the data obtained, we know that both inflammation and altered cerebral blood flow are the main candidates to justify the worse functional and vital prognosis of patients with acute stroke and SRBDs. Therefore, an area of interest would be to investigate anti-inflammatory treatments, particularly those aimed at inhibiting interleukin-6 and matrix metalloproteinase-9.

Regarding the role of CPAP, we can hypothesize that its use should eliminate the negative pressure effect in the thorax and thus normalize cerebral blood flow. However, this theory has not been confirmed by previous studies, which also suggest that there may be a pre-existing dysautonomic alteration due to chronic intermittent hypoxia that would not be quickly corrected with CPAP use [19,20]. Therefore, the timing of introducing CPAP therapy is crucial and needs to be studied. It is likely necessary to identify patients at risk of stroke who should be treated with CPAP to prevent developing a stroke, and if it occurs, to evaluate whether if outcomes are better. It is also essential to study the possibility of initiating SRBD therapy from the very first moment of the patient’s sleep after the onset of stroke, as the first hours after stroke onset are critical and when treatments have the most significant impact on future outcomes.

It is also important to conduct studies classifying strokes according to their different types, first dividing them into hemorrhagic and ischemic strokes, and within the latter, investigating whether there are differences between lacunar and non-lacunar ischemic stroke patients. This is due to the fact that the pathophysiology, prognosis, and clinical features of lacunar ischemic strokes differ from all other ischemic stroke subtypes [55]. It is also necessary to study the use of other devices different from CPAP, as we know that within sleep-related breathing disorders (SRBD), we also have central apnea disorders, which are not treated with CPAP. In this case, adaptive servo-ventilation could play an important role.

## Figures and Tables

**Figure 1 cimb-47-00138-f001:**
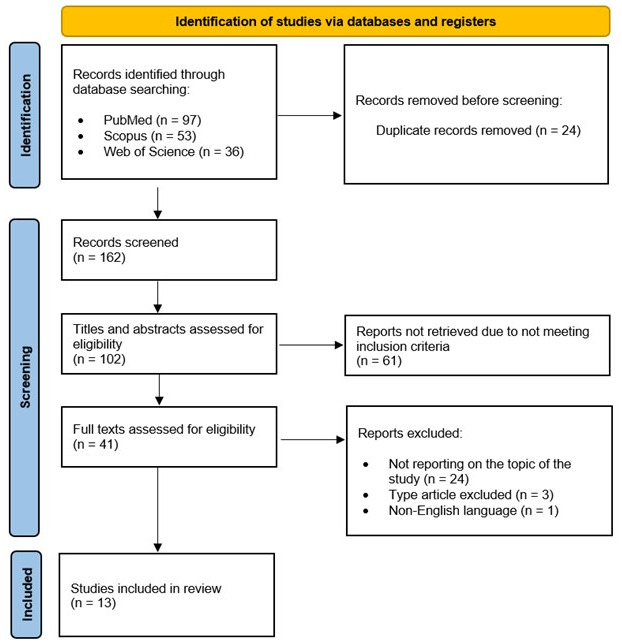
Preferred reporting items for systematic reviews and meta-analyses extension for scoping reviews (PRISMA-ScR) diagram for included studies [28,29].

**Table 1 cimb-47-00138-t001:** Summary table of included studies.

References	Study Type	No. Participants or Reported Case	Factors Assessed (Methodology)	Summary of Outcomes	Conclusions
Lévy et al., 2009 [35]	Literature review study	96 articles	- Review of overall studies on mechanisms of atherosclerosis - Data applicable to humans, animal models, and the therapeutic perspectives	Key factors for promoting atherosclerosis were: - Intermittent hypoxia (IH) - Blood pressure alterations - Hemodynamic strains on the vascular wall - Impairment in vascular reactivity - Lipid metabolism dysregulation - Activation of proinflammatory transcription factors at the vascular wall level	There are animal and clinical data supporting the role of IH in triggering aterosclerosis related to stroke.
Fei et al., 2021 [38]	Animal model study	159 mice	- Animal study on neuronal damage induced by OSA - Mechanisms by which intracerebral hemorrhage (ICH)-induced early brain injury regulates neural apoptosis	OSA induction produced: - Decreased survival rate, neurological score and neuron survival - Upregulation of protein expression levels of Caspase-3, Bax, cytokines IL-1β, IL-6, TNF-α and NF-κB - OSA-mediated induction of apoptosis and neuroinflammation-aggravated neuronal death following ICH	- OSA induced apoptosis and neuroinflammation aggravating neuronal death following ICH. - The molecular mechanism was partly dependent on the activating transcription factor/CHOP pathway.
Kunz et al., 2011 [40]	Population prospective study	76 patients	Comparison of serum inflammatory biomarkers in patients with acute stroke and OSA vs. stroke patients without OSA, including: - Soluble tumor necrosis factor receptor-1 and -2 (sTNF-R1 and sTNF-R2) - Tumor necrosis factor-β (TNF-β) - Soluble intercellular cell adhesion molecule-1 (sICAM-1) - Soluble vascular cell adhesion molecule-1 (sVCAM-1)	In patients with apnea-hypopnea index (AHI) ≥ 10/h: - sTNF-R1 and sTNF-R2 levels were significantly higher. - TNF-β, sICAM-1 and sVCAM-1. showed no significant differences.	- sTNF-R1 and sTNF-R2 may be part of the pathophysiological pathway linking OSA to stroke.
Medeiros et al., 2012 [42]	Cross-sectional prospective study	31 participants	Comparison of serum inflammatory biomarkers in patients with acute stroke (first week) OSA vs. stroke patients without OSA vs. controls, including: - Interleukin-6(IL-6) - Interleukin-1β (IL-1β) - TNF-α	- IL-6 increased in stroke patients with SA. - IL-6 is correlated with oxyhemoglobin desaturation and with desaturation index	Inflammation is an important unifying mechanism between OSA and stroke.
Ifergane et al., 2016 [30]	Cross-sectional prospective study	43 patients	Comparison of serum inflammatory biomarkers in patients with acute stroke and OSA vs. stroke patients without OSA, including: - IL-6 - TNF - Plasminogen activator inhibitor-1 (PAI-1)	- Levels of all 3 biomarkers were higher among patients with AHI ≥ 15	OSA was associated with significantly increased levels of inflammatory biomarkers
Yeh et al., 2017 [34]	Cross-sectional prospective study	20 patients	Comparison of inflammatory and oxidative stress biomarkers, sleep awakenings (messured by acelerometer) and rehabilitation outcomes in subacute stroke patients, including: - sICAM-1 - Glutathione peroxidase (GPx) - Malondialdehyde (MDA)	Positive correlations were observed between baseline level of sICAM-1 and number of awakenings at post-treatment (ρ = 0.51, *p* < 0.05).	- Patients with higher baseline levels of inflammation had higher numbers of awakenings at rehabilitation post-treatment.
Tsai et al., 2010 [37]	Literature review study	70 articles	Review of cognitive and neurobehavioral sequelae in patients with stroke and OSA	- AHI is the most common measure of OSA and is variably related to stroke and cognitive impairment. - There is significant overlap in the neuropsychological profiles of OSA and stroke patients.	- Sleepiness is the most common neurobehavioral sequela in patients with OSA and stroke. - CPAP treatment for as little as 2 weeks improves daytime sleepiness in most controlled trials
Chen et al., 2015 [31]	Cross-sectional prospective study	72 patients	Comparison of serum and urine biomarkers of oxidative stress and inflammation in patients with stable ischemic stroke and OSA vs. without OSA, including: - Serum C-reactive protein (CRP) - Serum IL-6 - Serum total antioxidant capacity (TAC) - Urinary 8-hydroxy-2-deoxyguanosine	- Participants with AHI > 30 had a significantly lower level of CRP. - TAC levels were significantly and negatively correlated with mean SaO2 levels.	- Adaptive antioxidative response to hypoxia emerges, and the role of OSA with respect to inflammatory reaction is attenuated in ischemic stroke patients with OSA.
von Känel et al., 2013 [32]	Cross-sectional prospective study	51 participants	Comparison of prothrombotic markers in OSA (AHI ≥ 10) (before and after CPAP treatment) and controls, including: - Plasminogen activator inhibitor (PAI)-1 antigen - D-dimer - Von Willebrand factor (VWF) antigen - soluble tissue factor (sTF) antigen	- OSA patients showed lower mean and amplitude of D-dimer. - OSA patients showed higher mean of plasminogen activator inhibitor-1.	There may be altered day/night rhythm of some prothrombotic markers in OSA. - CPAP treatment did not affect day/night rhythm of prothrombotic marker.
Jensen et al., 2018 [33]	Cross-sectional prospective study	28 patients	Comparison of: - Cerebral blood flow (CBF) - Cerebral metabolic rate of oxygen (CMRO2) - Lactate serum levels - In OSA patients (before and after CPAP treatment) vs. controls.	- In hypoxia, CBF significantly increased and blood oxygen concentration decreasing in control group. - CBF was unchanged in OSA patients.	- OSA patients had a reduced CBF in response to hypoxia. - CPAP treatment normalized hypoxia response.
Gregori-Pla et al., 2018 [36]	Cross-sectional prospective study	78 patients	CBF measured at supine to 30° to supine position on bed, comparing - Non-treated OSA group - CPAP-treated OSA group - Control subjects	- Moderate and severe OSA did not recover CBF when patients were tilted back to the supine position. - CBF response was normalized after CPAP treatment.	- Moderate and severe patients with OSA have altered cerebral vasoreactivity related to OSA severity.
Azarbarzin et al., 2012 [41]	Cross-sectional prospective study	113 patients	- Pulse rate response to respiratory events (ΔHR) was measured in Coronary Arterial Disease (CAD) patients with OSA (AIH ≥ 15) comparing groups treated with CPAP or not. Primary outcomes were: - Repeat revascularization - Myocardial infarction - Stroke - Cardiovascular mortality	- CPAP reduced risk of outcomes while increasing progressively pulse rate response to respiratory events.	- There is a protective effect (including against stroke) of CPAP in patients with CAD and OSA modified by the ΔHR. - Patients with higher ΔHR exhibit greater cardiovascular benefit from CPAP.
Mashaqi et al., 2021 [39]	Literature review study	50 articles	- Review of MEDLINE database (PubMed) for publications on Matrix metalloproteinase-9 (MMP-9), OSA, and cardiovascular disease	MMP-9 levels are elevated in patients with: - Stroke - OSA - Systemic hypertension - Myocardial infarction - Postmyocardial infarction heart failure	- MMP-9 levels are positively correlated with stroke severity at 12, 24, and 48 h from the onset of stroke - MMP-9 levels positive correlated with the size of the brain infarcted area

## Data Availability

The data that support the findings of this study are available on request from the corresponding author. The data are not publicly available due to privacy or ethical restrictions.

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
