# Peer review of "From Molecular to Clinical Implications of Sleep-Related Breathing Disorders on the Treatment and Recovery of Acute Stroke: A Scoping Review"

_cimb, 2025, doi:10.3390/cimb47030138_

Round 1
Reviewer 1 Report
Comments and Suggestions for Authors
Dear authors, your study aims to give an overview of what is known about moleculat mechanisms related to SRBD and acute stroke.
Whereas the structure of the review is based on Prisma (good), your search strategy failed to include several patient studies looking at stroke and SRBD/SA/OSA. Even looking at the included reviews, these are mostly about OSA and cardiovascular disease and not directly stroke.
I think it is important that you keep to your main outcome (stroke). Looking at the broad aspect of cardiovascular disease as sleep apnea, your results and discussion is not clearly linked to stroke.
As for the article structure, the search strategies (table 1) for each data base are not necessaryly to be exposed in the article, these can be attached.
Table 2 brings all studies in no clear order, nor grouped by study design/type of study, nor by thematic (stroke, cardiovascular disease, clinical study, animal study..) what make s the ubderstanding quite confusing. Your 13 included articles list 4 clinical studies enrolling stroke patients, 5 evaluate OSA pateints and cardiovascular outcomes, and 4 reviews mixing OSA and CPAP outcome,, OSA CVdisease, animal and human studies, OSA and CAD, so it is not clear to me, which were your inclusion criteria. There are several reviews looking at OSA/SA and stroke, stroke sequelae, as well as systematic reviews which are not cited.
The table 2 might be improved in the visual output. Do not write complete sentences, group the studies by study design and outcomes, use the same terms for outcomes, in its actual form it is difficult to read and the different outcomes can not be easily combined.
Your discussion is just the listing of the summary of each included article, but should have confronations/aggregation of information etc. Conclusions should not repeat results but answer your main question of your objectives (mapping existing research on molecular (biomarkers) mechanisms for SRBD and stroke).
The thema is very important, I suggest to re-organize the data.
Comments on the Quality of English Language
English needs some punctual corrections.
Author Response
|
Comments 1: [Dear authors, your study aims to give an overview of what is known about moleculat mechanisms related to SRBD and acute stroke. Whereas the structure of the review is based on Prisma (good), your search strategy failed to include several patient studies looking at stroke and SRBD/SA/OSA. Even looking at the included reviews, these are mostly about OSA and cardiovascular disease and not directly stroke. I think it is important that you keep to your main outcome (stroke). Looking at the broad aspect of cardiovascular disease as sleep apnea, your results and discussion is not clearly linked to stroke.]
|
|
Response 1: We appreciate your valuable feedback on our study. We understand your concerns regarding the scope and focus of our review. We would like to clarify that our approach was intentionally narrow, focusing on studies that combine both clinical and molecular data related to SRBD and acute stroke. This specific focus explains the limited number of studies included in our review. We acknowledge that we had to extract relevant data from studies exploring OSA and its relationship with stroke in the context of various vascular diseases. This was necessary due to the scarcity of research specifically addressing molecular mechanisms in SRBD and acute stroke. In response to your comment, we have made modifications in our results and discussion sections to clearly emphasize that our primary focus is on acute stroke. We have strengthened the connections between our findings and acute stroke outcomes throughout the discussion. These changes should help clarify our study's focus and methodology while maintaining the integrity of our original aim to explore the molecular mechanisms related to SRBD and acute stroke. We believe these modifications will address your concerns while preserving the contribution of our review to the field.
|
|
Comments 2: [As for the article structure, the search strategies (table 1) for each data base are not necessarily to be exposed in the article, these can be attached.]
|
|
Response 2: Thank you very much for the help, we completely agree, we have removed Table 1 and have placed that content as appendix A |
|
Comments 3: [Table 2 brings all studies in no clear order, nor grouped by study design/type of study, nor by thematic (stroke, cardiovascular disease, clinical study, animal study..) what make s the ubderstanding quite confusing. Your 13 included articles list 4 clinical studies enrolling stroke patients, 5 evaluate OSA pateints and cardiovascular outcomes, and 4 reviews mixing OSA and CPAP outcome,, OSA CVdisease, animal and human studies, OSA and CAD, so it is not clear to me, which were your inclusion criteria. There are several reviews looking at OSA/SA and stroke, stroke sequelae, as well as systematic reviews which are not cited..]
|
|
Response 3: We agree with the comment. First, it is important to clarify the criteria used to select the studies: - Studies had to answer the following PICO question: What molecular, pathophysiological, and clinical repercussions will sleep-related breathing disorders (SRBD) have on acute stroke regarding its treatment and recovery? - Studies assessing molecular, clinical, and therapeutic repercussions of SRBD on acute stroke as primary or secondary outcome variables. - Published articles up to December 2023 - Languages: English, Spanish, French, or German - Studies involving human or animal participants were included. - Methodology: quantitative, qualitative, and mixed-methods studies to consider a broad catchment of the topic. - Letters to the editor, editorials, commentaries, published protocols, conference proceedings, studies only published in abstract form, and case reports were excluded. On the other hand, we agree that the table 2 did not follow a logical order for easy comprehension; we have now reordered the table 2, starting with studies that include acute strokes in animals, then studies of purely acute stroke in humans, followed by subacute stroke studies, then recovery from acute stroke, and finally studies that address the relationship of SRBD with factors that intervene in the acute phase of stroke. It is true that in this last group, studies were not specifically designed to evaluate the relationship with stroke, but we have considered them because their results, such as prothrombotic factors, are important when developing and treating an acute stroke. |
|
Comments 4: [The table 2 might be improved in the visual output. Do not write complete sentences, group the studies by study design and outcomes, use the same terms for outcomes, in its actual form it is difficult to read and the different outcomes can not be easily combined. ]
|
|
Response 4: We agree with the comment. Table 2 has undergone significant improvements to enhance its clarity and usability. The studies have been reorganized and grouped starting with studies that include acute strokes in animals, then studies of purely acute stroke in humans, followed by subacute stroke studies, then recovery from acute stroke, and finally studies that address the relationship of SRBD with factors that intervene in the acute phase of stroke. This new structure provides a clearer overview of the evidence. Terminology for outcomes has been standardized across all studies, facilitating easier comparison. The outcomes themselves are now easier described, offering a more logical flow of information. Data presentation has been simplified through the use of abbreviations. These modifications collectively make the table more reader-friendly and allow for more straightforward synthesis of data across the various studies presented. |
|
Comments 5: [Your discussion is just the listing of the summary of each included article, but should have confronations/aggregation of information etc. Conclusions should not repeat results but answer your main question of your objectives (mapping existing research on molecular (biomarkers) mechanisms for SRBD and stroke). The thema is very important, I suggest to re-organize the data. ]
|
|
Response 5: We appreciate your feedback on our discussion section. We acknowledged that our previous approach lacked the necessary depth of analysis and synthesis. To address this, we restructured our discussion to focus on key themes and molecular mechanisms rather than summarizing individual studies. This provided a more cohesive overview of the relationship between SRBD and stroke.
We synthesized findings across studies, highlighting consistencies in the literature. This approach allowed for a more nuanced understanding of the molecular links between SRBD and stroke. Additionally, we incorporated a more in-depth analysis of the implications of these findings, discussing potential mechanisms and their relevance to clinical practice.
Finally, we revised our discussion to directly address our main research question regarding the mapping of existing research on molecular mechanisms and biomarkers for SRBD and stroke, without repeating results.
We thank you for helping us improve the quality of our manuscript. We recognized the importance of this topic and strived to present a more comprehensive and analytical discussion that better serves the scientific community. |
Reviewer 2 Report
Comments and Suggestions for Authors
The manuscript is describing a scoping review which tried to evaluate influence of SRBD on pathophysiology and outcomes of patients with ischemic stroke. It is clearly written, and literature search had some elements of systematicity, but substantial improvement are needed to make this manuscript acceptable and useful for the readership:
1. Although the authors insist on following Prisma-ScR reporting checklist, they are missing a few items from the list: the protocol was not pre-registered, and methodological quality /risk of bias of the included studies were not investigated (the latter is not mandatory).
2. Main problem is lack of whole picture of the problem that a reader can grasp; yet scoping reviews should provide exactly that. In the Discussion section, after short introducing paragraph, there is a chain of paragraphs, each one devoted to summary of one of the included studies. Where is a synthesis of all studies? Where is clear differentiation between hard outcomes of ischemic stroke and surrogate markers, which should give a reader clear picture of clinical relevance of the previous findings? Where is whole perspective of the problem? The authors should re-write the Discussion section and make it synthetic and more palatable for average reader.
3. There is no need to include case of a patient in this review, it is not helpful for understanding main issues. Please delete.
Author Response
|
Comments 1: [1. Although the authors insist on following Prisma-ScR reporting checklist, they are missing a few items from the list: the protocol was not pre-registered, and methodological quality /risk of bias of the included studies were not investigated (the latter is not mandatory).
|
|
Response 1: Thank you for your feedback on our scoping review. We appreciate your attention to detail regarding the PRISMA-ScR reporting checklist. We acknowledge that in our initial submission, we did not mention the pre-registration of our protocol. We would like to clarify that we have since registered our protocol, and it now has an assigned DOI number. This oversight in reporting was unintentional, and we apologize for any confusion it may have caused.
Regarding the methodological quality and risk of bias assessment, as you correctly noted, this is not mandatory for scoping reviews according to the PRISMA-ScR guidelines. Given the nature of our review and its broad scope, we chose not to include this assessment. However, we recognize that this information could provide additional context for our findings.
We will update our manuscript to include the protocol registration information and clarify our approach to quality assessment. Thank you for bringing these points to our attention, as they will help improve the transparency and completeness of our reporting.
|
|
Comments 2: [2. Main problem is lack of whole picture of the problem that a reader can grasp; yet scoping reviews should provide exactly that. In the Discussion section, after short introducing paragraph, there is a chain of paragraphs, each one devoted to summary of one of the included studies. Where is a synthesis of all studies? Where is clear differentiation between hard outcomes of ischemic stroke and surrogate markers, which should give a reader clear picture of clinical relevance of the previous findings? Where is whole perspective of the problem? The authors should re-write the Discussion section and make it synthetic and more palatable for average reader.
|
|
Comments 3: [3. There is no need to include case of a patient in this review, it is not helpful for understanding main issues. Please delete]
|
|
Response 3: Agree. We have deleted. Thank you for your help. |
Reviewer 3 Report
Comments and Suggestions for Authors
I I am unconvinced that a scoping review is appropriate here, given that the link between sleep disorders and stroke, its prevention and therapeutic role is very well recognised, I would have thought that a Systematic Review with its additional analysis of the studies for factors such as Bias and the generation of ODs ratios, would be more appropriate. I think this is particularly the case as their Abstract states “The studies highlight the significant association between OSA and cardiovascular conditions, including hypertension, myocardial infarction and heart failure, as well as the presence of elevated inflammatory molecules in patients with OSA and acute stroke”. I would suggest that the authors consider this approach and/or have a more robust justification for their use of a Scoping Review.
Presentational issues:-
Both Table 1 and 2 are very challenging to the reader and it is essential that these Tables should reformatted. I would suggest that aspects of Table 3 in particular, such as Factors Assessed, Summary, Key Results need greater brevity or placed in an appendix.
Figure 1 is not necessary in this review-I believe it to be entirely superfluous.
The Abstract's comments about the need for ethical approval are unnecessary.
Presentational issues:-
Both Table 1 and 2 are very challenging to the reader and it is essential that these Tables should reformatted. I would suggest that aspects of Table 3 in particular, such as Factors Assessed, Summary, Key Results need greater brevity or placed placed in an appendix.
Figure 1 is not necessary in this review-I beleive it to be entirely superfluous.
The Abstract's comments about the need for ethical approval are unecessary.
Author Response
|
Comments 1: [I I am unconvinced that a scoping review is appropriate here, given that the link between sleep disorders and stroke, its prevention and therapeutic role is very well recognised, I would have thought that a Systematic Review with its additional analysis of the studies for factors such as Bias and the generation of ODs ratios, would be more appropriate. I think this is particularly the case as their Abstract states “The studies highlight the significant association between OSA and cardiovascular conditions, including hypertension, myocardial infarction and heart failure, as well as the presence of elevated inflammatory molecules in patients with OSA and acute stroke”. I would suggest that the authors consider this approach and/or have a more robust justification for their use of a Scoping Review.]
|
|
Response 1: We appreciate your thoughtful feedback and understand your concerns regarding the appropriateness of a scoping review for this topic. While we acknowledge that the link between sleep disorders and stroke is indeed well-recognized, we would like to clarify the specific focus and rationale behind our approach.
Our study aims to investigate the molecular mechanisms implicated in the relationship between acute stroke and sleep apnea, particularly in the acute phase of stroke. Despite the established association between these conditions, there is a significant gap in our understanding of the specific molecular pathways involved, especially those with direct clinical repercussions in the acute stroke setting.
The decision to conduct a scoping review rather than a systematic review was based on several factors:
1. Limited research: While the general link between sleep disorders and stroke is well-documented, studies focusing specifically on molecular mechanisms in the acute phase of stroke are scarce. This paucity of targeted research makes a systematic review challenging and potentially less informative.
2. Heterogeneity of available data: The existing studies on this topic vary widely in their methodologies, outcomes, and molecular markers examined. This heterogeneity makes it difficult to perform the kind of quantitative analysis typically associated with systematic reviews, such as generating odds ratios.
3. Exploratory nature: Our aim was to map the current landscape of research in this specific area, identifying gaps and potential directions for future studies. This aligns more closely with the objectives of a scoping review.
4. Broad perspective: We sought to include a wide range of molecular markers and potential mechanisms, which is more suited to the inclusive approach of a scoping review.
We acknowledge that our abstract may have overstated the breadth of our findings, and we will revise it to more accurately reflect the specific focus on molecular mechanisms in acute stroke.
While we believe that a scoping review remains the most appropriate approach for our specific research question, we appreciate your suggestion and will provide a more robust justification for our methodology insction discution of the revised manuscript. We will also consider conducting a systematic review as a future step, once more targeted research becomes available in this specific area.
Thank you for your valuable input, which will help us improve the clarity and justification of our approach.
|
|
Comments 2: [Both Table 1 and 2 are very challenging to the reader and it is essential that these Tables should reformatted. I would suggest that aspects of Table 3 in particular, such as Factors Assessed, Summary, Key Results need greater brevity or placed in an appendix.]
|
|
Response 2: Thank you for your feedback and suggestions regarding our manuscript. We appreciate your attention to detail and the opportunity to clarify and improve our presentation of data.
In response to your comments and those of another reviewer, we have made the following changes:
1. Table 1: As suggested by another reviewer, we have removed Table 1 from the main body of the manuscript. The information previously contained in this table has been transferred to an appendix. This change allows for a more streamlined presentation of our main findings while still providing access to the detailed information for interested readers.
2. Table 2: We have thoroughly revised the information in Table 2. The content has been rewritten, reorganized, and summarized to enhance comprehension. We believe these changes will make the data more accessible and easier to interpret for our readers.
3. We haven’t done a table 3, we belive that yor comentary is abut table 2.
We appreciate your guidance in helping us enhance the quality of our manuscript. |
|
Comments 3: [ Figure 1 is not necessary in this review-I believe it to be entirely superfluous.]
|
|
Response 3: Agree. We have deleted.
|
|
Comments 4: [The Abstract's comments about the need for ethical approval are unnecessary. ]
|
|
Response 4: We appreciate your feedback regarding the Abstract section of our manuscript. We have deleted that information. |
Round 2
Reviewer 2 Report
Comments and Suggestions for Authors
The authors corrected the manuscript according to suggestions from the previous review. The manuscript is now sufficiently improved to merit acceptance.
Reviewer 3 Report
Comments and Suggestions for Authors
I am now satisfied with the justification, approach and presentation of this paper